# Salt Stress in Plants and Mitigation Approaches

**DOI:** 10.3390/plants11060717

**Published:** 2022-03-08

**Authors:** Gabrijel Ondrasek, Santosha Rathod, Kallakeri Kannappa Manohara, Channappa Gireesh, Madhyavenkatapura Siddaiah Anantha, Akshay Sureshrao Sakhare, Brajendra Parmar, Brahamdeo Kumar Yadav, Nirmala Bandumula, Farzana Raihan, Anna Zielińska-Chmielewska, Cristian Meriño-Gergichevich, Marjorie Reyes-Díaz, Amanullah Khan, Olga Panfilova, Alex Seguel Fuentealba, Sebastián Meier Romero, Beithou Nabil, Chunpeng (Craig) Wan, Jonti Shepherd, Jelena Horvatinec

**Affiliations:** 1Faculty of Agriculture, The University of Zagreb, Svetosimunska c. 25, 10000 Zagreb, Croatia; jonti.js@gmail.com (J.S.); jhorvatinec@agr.hr (J.H.); 2ICAR—Indian Institute of Rice Research, Hyderabad 500030, India; santosha.rathod@icar.gov.in (S.R.); giri09@gmail.com (C.G.); anugenes@gmail.com (M.S.A.); sakhare.akshaya@gmail.com (A.S.S.); birju1973@gmail.com (B.P.); bnirmaladrr@gmail.com (N.B.); 3ICAR—Central Coastal Agricultural Research Institute, Ela, Old Goa 403402, India; manohar.gpb@gmail.com; 4ICAR—Krishi Vigyan Kendra, KVK, Balumath, Latehar 829202, India; bd2511@gmail.com; 5Department of Forestry and Environmental Sciences, Shahjalal University of Science and Technology, Sylhet 3114, Bangladesh; fraihan-for@sust.edu; 6Department of Business Activity and Economic Policy, Institute of Economics, Poznań University of Economics and Business, Al. Niepodległości 10, 61-875 Poznań, Poland; anna.zielinska-chmielewska@ue.poznan.pl; 7Center of Plant, Soil Interaction and Natural Resources Biotechnology, Scientific and Technological Bioresource Nucleus, Universidad de La Frontera, Temuco 4780000, Chile; cristian.merino@ufrontera.cl; 8Departamento de Ciencias Químicas y Recursos Naturales, Facultad de Ingeniería y Ciencias, Universidad de La Frontera, Temuco 4780000, Chile; marjorie.reyes@ufrontera.cl; 9Department of Agronomy, Faculty of Crop Production Sciences, The University of Agriculture, Peshawar 25130, Pakistan; amanullah@aup.edu.pk; 10Russian Research Institute of Fruit Crop Breeding (VNIISPK), 302530 Zhilina, Orel District, Orel Region, Russia; us@vniispk.ru; 11Departamento de Ciencias Agronómicas y Recursos Naturales, Facultad de Ciencias Agropecuarias y Forestales, Universidad de La Frontera, Temuco 4780000, Chile; alex.seguel@ufrontera.cl; 12Instituto de Investigaciones Agropecuarias INIA, Carillanca, Temuco 8320000, Chile; sebastian.meier@inia.cl; 13Mechanical and Industrial Engineering Department, Applied Science Private University, Amman 11931, Jordan; beithounabil@yahoo.com; 14Jiangxi Key Laboratory for Postharvest Technology and Nondestructive Testing of Fruits & Vegetables, College of Agronomy, Jiangxi Agricultural University, Nanchang 330045, China; chunpengwan@jxau.edu.cn

**Keywords:** salt stress, neutral and alkaline salinity, plant–microbe associations, salinity and nanotechnology, soil amendments, artificial intelligence

## Abstract

Salinization of soils and freshwater resources by natural processes and/or human activities has become an increasing issue that affects environmental services and socioeconomic relations. In addition, salinization jeopardizes agroecosystems, inducing salt stress in most cultivated plants (nutrient deficiency, pH and oxidative stress, biomass reduction), and directly affects the quality and quantity of food production. Depending on the type of salt/stress (alkaline or pH-neutral), specific approaches and solutions should be applied to ameliorate the situation on-site. Various agro-hydrotechnical (soil and water conservation, reduced tillage, mulching, rainwater harvesting, irrigation and drainage, control of seawater intrusion), biological (agroforestry, multi-cropping, cultivation of salt-resistant species, bacterial inoculation, promotion of mycorrhiza, grafting with salt-resistant rootstocks), chemical (application of organic and mineral amendments, phytohormones), bio-ecological (breeding, desalination, application of nano-based products, seed biopriming), and/or institutional solutions (salinity monitoring, integrated national and regional strategies) are very effective against salinity/salt stress and numerous other constraints. Advances in computer science (artificial intelligence, machine learning) provide rapid predictions of salinization processes from the field to the global scale, under numerous scenarios, including climate change. Thus, these results represent a comprehensive outcome and tool for a multidisciplinary approach to protect and control salinization, minimizing damages caused by salt stress.

## 1. Introduction

Increased concentrations of dissolved salts (ions) either in (i) water resources used for (fert)irrigational purposes (electrical conductivity in water: EC_ir_ > 1.0 mS/cm) or in (ii) soil solutions/extracts (electrical conductivity of saturated soil extracts: EC_e_ > 2 mS/cm) usually induce one of the most widespread abiotic disorders in cultivated and native plant species, known as salt stress, after a mid-term period (e.g., one–two weeks) of exposure [1,2,3]. It is predicted that salt stress and related negative environmental implications will become even more critical, notably due to ongoing global climate change (e.g., more frequent and pronounced drought periods coupled with heat-stresses, underpinned evapotranspiration demands, over-increased average air temperature, rising sea levels, and the increasing tendency of generating grey waters, which are not purified to the appropriate level and are (re)used [4]. Accordingly, the most recent projections forecast the increase in irrigated agriculture from ~20% of totally cultivated land areas to 47% by 2030 [5], soliciting a difficult challenge to accomplish within a short period of time due to intense competition in the agri–domestic–industry triangle, posing markedly depleted and quality-constrained blue/green water resources [6].

Salt stress in agroecosystems disturbs crop food/feed yield production and quality due to a wide range of primary (osmotic stress, reduced nutrient uptake and growth) and more complex secondary salt-induced physiological disbalances (generation of reactive oxygen species and radicals which can damage proteins, membrane lipids, carbohydrates, and DNA structures) [1,3,7]. However, over-prolonged durations of salinity in permanent and unrecoverable soil degradation scenarios (e.g., dispersion of soil stable aggregates and structures, soil crusting, swamping, desertification) are realistic options depending on the geo-hydro-morphological and climatological conditions of salinity-exposed areas [2,8]. Based on the electrical conductivity of soil extracts (EC_e_), its pH_H2O_ reaction and exchangeable sodium percentage (ESP) index, across terrestrial ecosystems, the most relevant are three common types of soil salinity [9]:Saline soils (EC_e_ > 4 mS/cm, pH_H2O_ < 8.5, and ESP < 15)Saline-alkaline or saline-sodic (EC_e_ > 4 mS/cm, pH_H2O_ < 8.5, and ESP > 15)Alkaline or sodic soils (EC_e_ > 4 mS/cm, pH_H2O_ > 8.5, and ESP > 15).

It was confirmed that environmental implications could vary markedly among plant species and pedospheric conditions depending on salinity levels and intensity (duration). For instance, sodium chloride-induced salt stress, as one of the most common and widely elaborated (in controlled and natural conditions) abiotic stresses [10], is caused by relatively neutral salts, i.e., NaCl dissociation in the water matrix generates a neutral pH reaction (pH = 6.998; Section 2). In contrast to this salt type, in sodium carbonate (Na_2_CO_3_)-induced or potassium carbonate (K_2_CO_3_)-induced salt stress conditions, the dissolution of a weak carbonic acid and strong sodium/potassium hydroxide will generate relatively more alkaline reactions (pH > 10.5; Section 2). Given the complexity of the environmental implications of a particular soil type, each salinity-induced disorder (stress) will impose specific implication(s) and thus should be considered and adequately managed. 

## 2. Neutral and Alkaline Salinity and Impacts to Plants 

Table 1 presents some of the most relevant scenarios of salt-stressed rhizosphere conditions, both with neutral and alkaline salt types, based on the biogeochemical speciation approach. Briefly, particular salt types were separately dissolved in the rhizosphere solution, which corresponds to wide uncontaminated mineral soil conditions [11]. 

According to the modeling results, the calculated pH values were shifted to mid-alkaline reactions (7.9–8.0), with confirmed organo-complexation of Mg (approximately 10%) and Ca (approximately 6%) within the dissolved organic pools. In addition, predicted mineral precipitations were attributed mostly to Ca/Mg phosphate and carbonate minerals (e.g., brucite, dolomite, hydroxyapatite, huntite, vaterite; Table 1).

Both salt types, neutral and alkaline, in certain scenarios can induce relevant salt disorders, i.e., stresses. However, the stress induced from neutral salt (NaCl, MgCl_2_, Na_2_SO_4_, CaCl_2_) will greatly differ from that induced by alkaline salt (Na_2_CO_3_, NaHCO_3_, K_2_CO_3_) [1,13] (Table 1). Accordingly, it was shown that more neutral salts mostly disrupt macro/micronutrient homeostasis, causing adverse osmotic imbalances and damage [14]. In the presence of more alkaline salts, the negatively induced effects will be almost identical; however, additional adverse impacts will be further aggravated due to the increased (alkaline) pH reaction of surrounding media, different ionic strengths, and different biogeochemical reactions, along with chemical speciation (Table 1). For instance, under more alkaline rhizosphere conditions, the mobility and availability of certain essential nutrient chemical forms (free Ca, Mg, Cl, H_2_PO_4_^−^) can be markedly reduced because of precipitation reactions and consequently homeostasis imbalance [13,14,15] (Table 1). Furthermore, a high alkaline pH reaction can additionally damage the structure of the root cell membrane, disrupting its structural integrity and functionality [1,16,17] (Figure 1A–C). Additionally, significantly lower tolerances of plants to alkaline vs. neutral salt stress have been documented thus far [17,18,19].

However, it was shown that excessive salinity can impose crucial implications on trace element soil biogeochemistry (Table 1), consequently either improving or limiting the uptake and accumulation of trace elements. For instance, the authors of [20] recently detected that NaCl salinity-induced root exudates in halophytic mangrove plant species (*Avicennia marina*) are very effective in binding Cu^2+^, Mn^2+^, and Cd^2+^, which can limit not only the phytoavailability but also the transfer of metals in deeper aquatic systems. Additionally, the authors of [21] confirmed inhibited Cd uptake in two other mangrove species (*Rhizophora apiculata* and *Avicennia alba*) exposed to increased salinity. In contrast, it was confirmed that salinity can enhance the uptake and deposition of toxic Cd and/or essential phytonutrients (Cu, Zn, Mn) in the native halophyte *Carpobrotus rossii* [22]. Similar effects were also observed in different glycophytes, such as in edible amaranth cultivars [23,24], muskmelon [25], radish cultivars [10], and strawberry (Figure 1). Such implications can be explained by geochemical interrelations in the rhizosphere. Namely, increased ionic strength in the rhizosphere solution under the presence of dissolved Cl^−^ can enhance trace element mobility (Table 1) via complexation reactions, accompanied by saturated reactive sites of the soil matrix through adsorption with Na^+^ [11]. 

To cope with excessive salt forms (Table 1) and different alkalinity- or salinity-induced disorders (Figure 1), higher plants, notably tolerant to excessive salinity, have developed a wide range of abiotic stress-adaptive strategies, including detoxification, regulating osmotic adjustment, maintaining cationic/anionic balance, scavenging reactive oxygen species, and synthesizing compatible solutes [13,28,29]. Unfortunately, most halophytes are still not as relevant as agricultural food production, although some, such as mangroves, represent irreplaceable ecological value in coastal mariculture as well as environmental protection from different anthropogenic and natural pressures (pollutions, salinization [20,21]), assured by invaluable genetic pools for possible bioengineering developments (Section 3). 

## 3. Sustainable Approaches and Solutions to Improve Plant Nutrition and Crop Production in Saline Conditions

It is possible to implement a wide range of sustainable preventive and proactive (reclamation) approaches separately and/or in combination to improve plant salt resistance and crop nutrition under salt-affected conditions (e.g., Figure 2). For instance, some halophytic strategies (traits) could be transferred to glycophytes (most cultivated crops, relatively sensitive to salinity), improving their resistance to salinity. Breeding and genetic approaches, such as the selection and creation of salt-resistant genotype(s), over (i) traditional breeding processes [30], (ii) marker-assisted selection [29,30,31,32], (iii) molecular and transgenic approaches [33], or (iv) genome editing (using the CRISPR/Cas9 tool) [34], have been the focus for an extended time, and some solutions have been successfully implemented for the alleviation of salt-affected crop production. However, certain constraints, such as high technological dependency, time-consuming procedures, unpredictable genetic gain, and extraordinarily diverse genotype–environment interactions (multi-collinearity [33]), still represent substantial limitation(s) in the progressive improvement of these approaches to increase salt tolerance in target crops. 

Application and adaptation of specific agro/technical/technological operations can also ensure a wide spectrum of land, water, and crop management solutions for controlling and avoiding detrimental effects of salinity to crops [12]. Some of the most applicative strategies are controlled water management over the applications of (i) modern, low-pressurized, and localized irrigation [4], and if necessary, (ii) surface/underground drainage systems [35]. Both systems can help maintain salinized groundwater levels below the critical root zone level and leach concentrated salts from the rhizosphere. 

Spatiotemporal adaptation of cropping patterns (e.g., during high evapotranspiration demands, on the lowest terrain positions), such as growing salt-resistant crops/cultivars/varieties in a single, double, or multiple cropping system (agroforestry, combining forage and cereal crops), has been confirmed as a very effective option to alleviate soil salinity and co-occurring environmental constraints of (semi)arid (agro)ecosystems (Figure 2). For instance, mixed cropping (vs. mono-cropping) with two or more cultures simultaneously has been confirmed to be more effective in soil C and N restoration [36]. The same authors reported that mixed cropping demonstrated changing effects on crop growth, which depends upon the plant species. It also offers better protection against soil deterioration and the disruption of pests and pathogenic bacteria and fungus. In addition, this system helps to reduce water erosion and groundwater salinization/contamination. In many developing countries, intercropping has been effectively used in low-input harvesting systems for enhancing land use and cultivating water and nutrient regimes. In Bangladesh, farmers have introduced intercropping systems into sugarcane farming, which increases the midterm return (in 12–14 months) along with the total profit [37]. This mechanism boasts improved water uptake, better root absorptions, or balancing exploration over the soil profile. Salt-affected areas often overlap with water-stressed, organically depleted, and poorly structured sandy soils, which are knowingly compatible for implementation in land and water conservation practices (humification, reduced/zero tillage) and can additionally underpin crop nutrition under saline conditions (Figure 2). 

Grafted crops, which combine within- or inter-species organisms at the rootstock and scion, are widely used in horticulture mostly to overcome different constraints and stresses, including salt stress. It was confirmed that grafting scions with more salt-tolerant rootstock varieties can improve the salt resistance of grafted plants over restrictions in salt (Na^+^, Cl^−^) uptake and enhance antioxidant enzymatic activities and hormonal adaptations to saline environments [38,39]. Grafting among genetically different and distant genera (species) is still widely unexplored but seems to be a very promising approach to improve many physiological reactions, such as salt-induced ones [40]. 

Soil amelioration with certain organic (more in Section 4) and/or inorganic (lime, gypsum, bottom and fly biomass/coal ash, saturated mud from sugar refineries) conditioners and amendments can also reclaim constrained lands and beneficially enhance crop production in such conditions in multiple ways. For example, applying an appropriate drainage system and providing quality irrigation water for leaching accumulated salts can be an effective measure for reclamation of saline soils. Still, saline-sodic and sodic soils (with elevated levels of Na^+^) usually demand the application of appropriate (Ca-/Mg-based) amendments to aid the amelioration [9,12]. Additionally, both Ca and Mg have the potential to further alleviate (sub)soil constraints over (i) stabilizing and improving the soil structure, (ii) reducing the sodium adsorption ratio (SAR), (iii) increasing flocculation, through improving soil structure, and thus decreasing clay dispersion, and (iv) improving water–air relations and many others (Figure 2).

Exogenous application of phytohormones has also been confirmed as a very effective and promising strategy against environmental stresses, including salt stress [41]. Phytohormone imbalance is very often a salt-induced phytoreaction. Phytohormones are a broad group of naturally occurring molecules or compounds (e.g., ethylene, auxin, gibberellins, cytokinins, strigolactones, brassinosteroids, abscisic/jasmonic/salicylic acid) which regulate plant growth and development under homeostasis and are irreplaceable in signaling transduction pathways during reaction stresses [42,43]. Recent studies suggest how phytohormones might be a crucial metabolic engineering target for creating salt stress-tolerant crop plants (see review in [43]). For instance, auxin can improve the growth performance in plants under salinity stress [44], albeit in some crops, salinity can reduce auxin levels. Moreover, exogenous addition of salicylic acid can effectively increase endogenous auxin and abscisic acid content and improve the growth performance in salt-stressed corn plants [45]. Similarly, the exogenous addition of jasmonic acid also seems to have the potential to alleviate salt-induced adverse impacts in plants. The latter is related to improved physiological properties (e.g., increase chlorophyll content and antioxidant enzymatic activity, reduce lipid peroxidation, improved K nutrition), which enhance plant growth and yield performance [41,46,47].

## 4. Salinity Amelioration by Organic Amendments 

New research strategies that promote the benefits of different organic amendments for plant growth in saline/sodic soils report about the reduction of oxidative and osmotic stress, improving the conductance and stomatal density and the seed germination rate, prompting an increase in microbial activities [48], and many others. Implementing organic materials demonstrated significant benefits, improving the saline soil biome by enriching it with compost, green manure, poultry manure, and sugarcane remnants (press-mud) [49,50]. These organic amendments heighten the dissolution percentage of calcite (CaCO_3_) via the increased formation of carbonic acid while improving the binding of the small particles, effectively forming substantially sizable aggregates that remain unwavering within water [51]. This method is effective in both calcareous as well as non-calcareous soils because the large-sized individual organic particles create channels in poorly structured saline or sodic soils, and thus aide in ameliorating the soil permeability while leaching Na^+^ from the cation exchange sites over the soil profile [52]. The selection of a sustainable reclamation technique and organic material is an extremely important factor that should be determined via the analysis of both site-specific geographical and soil physicochemical parameters [53]. Among a wide range of soil organic amendments, biochar has been intensively studied recently as effectively improving the physicochemical and biological properties of saline/sodic soils. 

Identical to non-saline soils, salt-affected soils benefit from the addition of biochar due to the freshly provided habitat created from the biochar, encompassing the ability to sustain vast multitudes of soil microorganisms, providing essential living elements to be compounded with the gained organic carbons and nutrients. Moreover, biochar stabilizes the soil structure, enhancing physical properties by balancing both the air porosity and water content, in relation to the cation ion exchange capacity [49]. Average types of biochar’s will increase the rate at which salt leaching occurs, effectively remediating the site for the immediate use of crop farming. Additionally, soil organic carbons aide in binding soil aggregates for a sustained long-term capacity in comparison to some other types of organic amendments stemming from non-degradable molecular makeups [53]. Biochar application improves the total porosity and water-holding capacity of salt-affected soils, but the effect appears to depend primarily on the feedstock type in combination with the organic material that is being used as the base source for the final product [52]. The reason for this is because biochar is created via the burning of organic materials in conditions either lacking or without O_2_, presenting a product that is a C-rich material achieved utilizing temperatures ranging from 300 to 1000 °C [54]. Due to the different organic constitutes that biochar is composed of, not all types will expend influences similar to that of one particular soil type, as well as no individual biochar can be effective within all (saline) soils [55]. This can be explained by examining biochar that is created utilizing non-woody raw materials, such as plant residues and numerous types of manure that are ample in nutrient content while rendering a less stable C with a higher pH than biochar generated from dry plant mass [56]. Thus, using biochar as a soil amendment in saline soils will effectively ameliorate the soil profile for superior growing conditions as various studies have proven the application in mitigating damages caused by salt stress [57]. Beneficial implications under biochar application are accomplished by: (i) the reduction of transient N via the process of adsorption, (ii) the release of both macro- and micro-mineral nutrients, and (iii) the decrease in stress factors caused by osmosis accomplished via improved water availability within the soil [58]. Due to strong absorptive properties, extremely high porosity, cation exchange capacity, and large surface area, biochar bind potentially toxic salt ions (Na^+^) at different magnitudes [56]. Moreover, such properties allow the desorption of potentially beneficial ions into the soil, effectively ameliorating nutrient misbalances caused by salinity [12,54].

Growth parameters such as photosynthetic rate, stomatal conductance, and transpiration rate are confidently influenced by biochar treatments, suggesting that biochar will reduce the adverse effects of salinity pressure on plants [52]. In addition, biochar can improve vital components related to crop yield, such as shoot biomass, root length, as well as yield in potatoes [56], maize, and tomatoes [59], grown under salinity conditions. The vast reaching impact of biochar on both the production of biomass and growth of herbaceous species can be examined in studies that have allowed *Prunella vulgaris* and *Abutilon theophrasti* to become exposed to salinity stress, revealing that biomass and plant growth were positively affected in both plant species in comparison to the control. However, it should be noted that biochar did not have a significant influence over the photosynthetic boundaries in either species while under salinity stress factors [60]. It can be stated that the response from each individual plant species differs enough to create prerequisites in order for a specific type of biochar to be recommended. Photosynthetic parameters increased within amended soils, leading to the rate of which plant growth stimulation occurred [61]. These findings revealed that biochar is able to be utilized as a stable organic amendment to soils for the purpose of mitigating salinity in grain crops [56]. The primary reason was that the utilization of biochar that has been tested for each area of specific soils and crop types has shown that the reduction in water was due to induced stomatal closure and regulation of transpiration, causing a higher efficiency [48], and thus leading to the preservation of both water balance and leaf turgidity within saline soil biomes. Plants develop antioxidant defense systems to cope with salt stress induced by oxidative damage. In addition, it has been confirmed that the increase in antioxidant enzymes triggered by the activation of plant defense mechanisms can be regulated by the application of biochar [58]. At heightened natural salinity levels (EC 1.26–2.00 mmhos/cm), it was recorded that lower catalase (CAT) and peroxidase (POD) activity occurred within biochar treatments of a 5% capacity, while lower superoxide dismutase (SOD) activity was recorded at treatment capacities of 2.5% accordingly [54]. However, at the biochar dosage of 10% (vs. control), a nonsignificant impact on antioxidant enzymatic activities was recorded [54]. Thus, a small percentage of biochar amended into the soil can alleviate many of the salinity-induced harmful effects on antioxidant enzymes. However, at higher (>10%) biochar application rates, negative consequences related to the increase in antioxidant enzymes [59] could be expected, the main reason being that there is a negative impact from the addition of biochar on growth due to high salinity and N immobilization [60]. Overall, the application of biochar reduced plant Na uptake due to transient Na+ binding, again due to its high adsorption capacity, which is responsible for decreasing osmotic stress by enhancing the soil’s moisture content and releasing mineral nutrients into the soil solution [56]. This point to the improved K/Na ratio, through which enhancing potassium (K) availability will substantially increase the majority of grain type plant growth and yield under saline soil stress factors [58]. 

## 5. Salinity and Plant–Microbe Associations

Microbial communities present in the rhizosphere are influenced by soil chemical conditions (pH, salinity, organic matter content) and plant (root architecture and depth, rhizodeposition rate) interactions [11,62]. Associated symbiotic endobacteria and bacteria, mycorrhizal fungi (i.e., autotroph microbes), free-living microbial decomposers, and other soil heterotrophs exist in the rhizosphere [51,63]. It was confirmed that plant–microsymbiotic associations have crucial functions in plant nutrition [62], plant performance, resistance to (a)biotic stresses [64], and adverse environmental conditions (nutrient imbalances, injuries by pathogens, soil acidity/alkalinity), while in return, microbes profit from assimilated plant C supply [64]. For example, on average, ~½ of net primary production (photosynthetically assimilated C) is translocated from the shoot-to-rhizosphere, out of which ~50% is retained in the root, >30% is spent as autotrophic (root + endomicrobial) respiration, and the remaining >15% become soil organic rhizodeposits [50,65]. 

It was found that soil salinity (NaCl) alone or in combination with other abiotic stresses (metal toxicity, alkalinity, water stress) can suppress certain plant–microbial associations and their population activity, and organic rhizodeposition [10,11,62]. Metabolic profiles of the root, rhizosphere, and root exudates can be markedly compromised in response to NaCl exposure and can differ among plant cultivars [24]. Furthermore, the authors of [66] confirmed antagonistic interrelations between soil salinity and microbial biomass C, concluding that salinity induced a negative impact on microbe biomass/activity. However, plant–microbe interrelations, notably with particular symbiotic-associated bacteria (e.g., N_2_-fixing) and/or arbuscular mycorrhizal fungi (AMF), have been confirmed as up-and-coming options for mitigating salt stress effects in plant species either sensitive or tolerant to saline environments [67,68,69]. 

### 5.1. Salinity and Symbiotic Bacterial Associations

One of the critical naturally relevant rhizosphere–microbe associations for mitigating salt stress involves specific N-fixing-associated bacteria (AB) groups, which act as plant growth promoters as well. Some AB growth enhancers improve salinity tolerance by generating specific enzymes (e.g., 1-aminocyclopropane-1-carboxylate deaminase), metal-organic complexes (e.g., siderophores), and hormones, fixing atmospheric N_2_ and solubilizing fewer mobile phosphates to more bioavailable forms [69]. Only two types of symbioses where N_2_-fixing soil bacteria induce physical nodular connections with the root of associated host plants have been recognized to date: (i) rhizobia and (ii) actinorhizal plant-bacterial symbioses [70]. In the former association, *Rhizobia* (Gram–) creates a symbiosis with approximately 80% of legumes and some *Ulmacea* spp. [71,72]. In contrast, the latter is an association between actinobacteria of the genus *Frankia* and mostly woody plants as hosts [70]. However, in both associations, the microsymbionts generate specific new anatomical parts (nodules on the root interface of its host), where atmospheric N_2_ is fixed and photosynthetically fixed carbohydrates are being supplied by the host plant [72]. In studies with different *Rhizobium* species/strains (e.g., *Rhizobium* spp. strain AC-1/AC-2, *Rhizobium* PMA63/1, *Rhizobium tropici* CIAT899, *Rhizobium* strain USDA 208), it was confirmed that symbiotic association in plants exposed to salinity stress increased the dry weight of biomass in *Acacia nilotica* [73], *Acacia ampliceps* [74], bean [75], and soybean [76]. Such *Rhizobium*-induced improvements in host plants can primarily be attributed to more effective N_2_-fixing symbiosis given that acetylene reduction activities were confirmed at very high salinity concentrations [74]. Recently, the authors of [1] exposed *Rhizobium*-associated *Medicago sativa* plants to salt stress (200 mM NaHCO_3_) and found that greater salt resistance was accompanied by higher levels of antioxidants (SOD, POD, GSH), osmolytes (sugar, glycols, proline), organic acids (succinic, fumaric, and α-ketoglutaric acid), and metabolites activities (involved in N-fixation) than non-symbiotic alfalfa plants.

The second, naturally less occurring actinorhizal symbiosis, is performed among the *Frankia* genus (i.e., Gram+ filamentous actinobacteria) and approximately 260 so far confirmed plant species, which are primarily perennial species (*Betulaceae, Casuarinaceae, Myricaceae, Rosaceae, Eleagnaceae, Rhamnaceae, Datiscaceae,* and *Coriariaceae* [67]) and are not directly involved in crop food production, albeit they are important for the protection of saline semi/arid (agro)ecosystems. It was documented that *Frankia*-associated symbiosis enables host actinorhizal plants to grow in highly constrained soils (contaminated, water-deficient/logged, nutrient-deficient), including intensely saline/alkaline ones [77,78]. For example, in *Casuarina glauca* and *Casuarina equisetifolia* exposed to NaCl salinity (up to 500 mM), inoculation with *Frankia* strains CcI3 and CeD improved plant height by up to 66% and 45%, respectively, with significantly increased biomass (shoot, root, and total), dry weight, and proline and chlorophyll contents compared to uninoculated control plants [79]. The authors explained this result by improved N nutrition and photosynthesis potential in inoculated (vs. uninoculated) plants exposed to salt stress. Similar results were observed using *Frankia*-inoculated associations in *Alnus glutinosa* trees grown in alkaline and saline anthropogenic sediment [80].

Currently, most actinorhizal woody species (e.g., from the *Casuarinas* family) are largely exploited in land reclamation and restoration of mining, metal-contaminated, and salt-affected environments [81], followed by agroforestry, crop, and soil protection from wind and wildfire influence and erosion [79]. However, genotypic predispositions of actinorhizal plants and the associated actinobacteria *Frankia* represents valuable potential for further exploitation and improvement of cultivated crops and their symbiotic associations for food/feed production in salt-affected (agro)environments, although extremely complex processes must be elucidated by comparative genomics and proteomics prior to this. Accordingly, the authors of [78] revealed that 2/3 of salt/osmotic stress-resistant strains of *Frankia* (able to withstand extremely saline environments, 475–1000 mM NaCl) shared 153 single-copy genes (a central code portion for hypothetical proteins), hundreds of genes were differentially expressed under salt and/or osmotic stress, and up to 19 salt and/or osmotic stress-responsive proteins were detected.

### 5.2. Salinity and Symbiotic Fungal Associations

Arbuscular mycorrhizal fungi (AMF) belong to the phylum Glomeromycotan, one of the most important groups of soil microbes, which can establish symbiotic inoculation with the roots of over 80% of terrestrial plant species [69]. AMF are well established and naturally occurring microbiota of saline soils [82], and multiple beneficial implications for symbiotic-associated glyco/halophytic species were confirmed in different study types. For instance, it was shown that AMF (e.g., *Glomus claroideum, Glomus intraradices, Glomus macrocarpum, Glomus mosseae, Paraglomus occultum, Rhizophagus intraradices*) in associated plant species (e.g., olive, acacia and citrus trees, corn, *Sesbania aegyptiaca*) can mitigate salt stress and enhance plant growth over improvement of water absorption capacity, nutrient acquisition and uptake, accumulation of different osmoregulators (proline, betaines, polyamines, antioxidants) to adjust cell osmopotential, physiological processes (photosynthetic C assimilation, transpiration) and molecular performance [69,83,84,85,86]. In addition, other studies have shown that AMF colonization can reduce the uptake of Cl while simultaneously preventing Na translocation to shoot [83] or reducing Na and improving Mg uptake under salinity [86]. It was concluded that the numerous benefits of AMF to associated host plants exposed to saline impacts could be further improved by the selection of more efficient fungal strains [67].

## 6. Salinity and Nanotechnology-Based Solutions

The exploitation of nanotechnology-based solutions is growing rapidly in different spheres of human activities, including (agro)ecosystems, i.e., crop food production [87]. For instance, applications of certain nanomaterials (e.g., single-/multi-walled C-based nanotubes, polymeric chitosan, graphene, fullerol, fullerene), nanoparticles (nano-fertilizers, nano-pesticides), and nano-based technologies and approaches (nanofiltration of brackish and/or grey water resources for irrigated cropping, trace element transport, and deposition within crop tissues) have been shown to be very promising strategies and alternatives to alleviate nutrient disorders and enhance crop food production under different abiotic conditions, including excessive salinity and induced salt disorders [3,88,89,90,91,92]. 

Different nanoforms of metal oxides (ZnO, CuO, TiO_2_, CeO_2_, Fe_2_O_3_, Fe_3_O_4_) have been intensively studied as applied agrochemicals, i.e., phytonutrients [93], plant growth regulators [94,95], and pesticides [88]. For instance, it was shown that TiO_2_ nanoparticles (NPs) at certain levels can improve the seed germination and seedling growth of wheat [96] and spinach [97], which was likely attributed to their nano-sizes, enabling penetration into the seed during the treatment period and allowing the NPs to exert their enhancing functions during growth [97]. Additionally, growth promotion might be supported by the photosterilization and photogeneration of reactive oxygen radicals induced by TiO_2_-NPs, which additionally improved stress resistance and promoted capsule penetration for the H_2_O and O_2_ uptake needed for fast germination [88,97]. Next, it was revealed that multi-walled C nanotubes (at the concentration range of 10–40 mg/L) can significantly improve the seed germination and growth of tomato plants. It was hypothesized that such an outcome was due to the capability of C tubes to penetrate the seed coat and therefore promote water uptake [98], which is relevant for early growth stages in cropping of (semi)arid systems. Recently, the authors of [91] showed that two C-based nanomaterials (multi-walled C nanotubes and graphene NPs) added to a growing medium also significantly improved the seed germination rate and total biomass of switchgrass and sorghum plants. In the same study, it was confirmed that the addition of graphene or C-based nanotubes to NaCl-exposed (100 mM) growing medium significantly reduced symptoms of salt stress in test crops. This outcome was explained by (i) the impact of C-based NPs on the plant transcriptome performance (e.g., enhanced expression of aquaporins) and (ii) physical interactions of C-based NPs with toxic ions (i.e., by removal of toxic Na^+^ ions from salt solution). Water channels (aquaporins) are crucial for water uptake/transport, notably under stressful conditions, and it was confirmed that the overexpression of the wheat TaNIP aquaporin gene in transgenic *Arabidopsis* enhanced salt tolerance compared to wild-type plants [99], while the wheat TaAQP8 aquaporin gene improved salt tolerance in transgenic tobacco [100]. Next, the authors of [3] combined the inhibitor/scavenger test and genetic approach and documented that multi-walled C nanotubes enhanced salt tolerance in rapeseed seedlings exposed to NaCl stress. They pointed out how generated endogenous NO (as an important signaling molecule that can enhance salt tolerance in some plants) might act downstream of multi-walled C nanotubes, signaling salt tolerance against NaCl stress in plants, though the reestablishment of redox and ion homeostasis is required.

Numerous restrictions and relatively high demands for the application of some traditional agrochemicals might be an important comparative disadvantage with respect to nano-based applications, notably in semi/arid areas where water stress overlaps with salt stress and nutrient deficiencies. For example, some traditional fertilizers and conditioners (e.g., gypsum, zinc-sulphate) for the amelioration of saline/alkaline soils are still the most used agrochemicals globally, mitigating nutrient (e.g., Zn) deficiency and other pedo-constraints (explained in previous sections). However, some conventional fertilizer forms pose quite a low use efficiency (20–50%) [89], their application is costly/technically demanding, and they can stay inactive if they persist in salt (undissolved) form (e.g., water-deficient conditions), although fertilizers and conditioners can be very mobile, i.e., pollute certain natural resources (e.g., agrochemicals in runoff waters [50]). Nano-based phytonutrients have been confirmed to be more effective (vs. some conventional fertilizer salts) due to their specific mechanisms of action (i.e., increased active surface area), improved use efficiency due to slow and controlled release, decreased nutrient losses, and lower deterioration of the environment considering their lower application dosages [89,101]. Additionally, most phytonutrients can be incorporated into the nanostructures of naturally occurring zeolites, i.e., Si-Al minerals, with a huge active surface area and 10-fold higher cation exchange capacity than soil [89].

Although the complete beneficial mechanism of NPs to plant physiological processes is still not fully explained and clear, it was shown that in nano-based forms, some metal elements (Cu, Zn, Fe) are less toxic than their salt forms (e.g., Zn and Fe NPs by 30-fold and 40-fold, respectively, vs. their sulphate salt forms) [102]. However, the authors of [103] evaluated the impacts of metallic NPs (multi-walled C nanotubes, Al, alumina, Zn, and ZnO) in six test plants and observed that only Zn-NPs and ZnO-NPs markedly inhibited seed germination and root growth, indicating that the suppressed impact varied significantly depending on the NPs, plants, and applied concentration as well. Additionally, in some of the most recent studies, it was confirmed that ZnO-NPs (vs. conventional zinc-sulphate form) imposed a more positive impact on growth and physiology performance in coffee plants [101], while Fe_2_O_3_ nanoparticles are effective in replacing traditional Fe fertilizers in the cultivation of peanut crops in sandy pedospheres [104].

Some of the previously mentioned secondary induced disorders, as a consequence of (a)biotic stresses, are elevated levels of reactive O_2_ species (ROS), which stimulate lipid peroxidation reactions. However, some studies confirmed that certain micronutrients are even more effective in inhibiting ROS formation and/or antioxidant performance [102], inducing other beneficial impacts (multifunctional biostimulants) to plants if present in the form of NPs (vs. conventional commercial salts), making them novel and bio-safe nano-modulators, e.g., [29]. Similarly, it was confirmed that the addition of Si-NPs (1 mM SiO_2_) improved lentil genotypes’ germination and early seedling growth after exposure to 100 mM NaCl and stimulated immunity mechanisms against salt toxicity [105]. Additionally, foliar application of ZnO-NPs increased the chlorophyll content, quantum yield, and biomass production in NaCl salt-stressed sunflower cvs. to a higher extent than the application of the conventional ZnO form [106], while the authors of [107] found that ZnO-NPs can partly mitigate the effects of salt stress in different tomato cvs., mostly through the upregulation of superoxide dismutase and glutathione peroxidase activities. Similarly, the authors of [108] revealed that the application of Se-NPs can induce positive impacts in salt-stressed tomato crops (mostly over elevated photosynthetic pigments and improved the photosynthetic capacity), as well as improve the content of some beneficial biocompounds (e.g., lycopene, β-carotene, flavonoids, phenols) in tomato fruits.

The prolonged performance of nano-based fertilizers over their slow release might also be a very effective approach in alleviating crop macro/micronutrient deficiency, especially in rain feed saline agroecosystems (Figure 2). For instance, an incorporation of N:P:K fertilizers in the form of NPs of chitosan (cationic, biodegradable, bioabsorbable, and bactericidal polymers) can be obtained with different compounds and with various stability of their colloidal suspensions (higher with N and K than with P addition) [109]. Consequently, some nano-encapsulated fertilizer forms have been confirmed with better utility and efficiency in inoculated soils, together with their controlled and prolonged release (effective even 60 days after application) into the soil, e.g., [93,110].

In addition, negative (sometimes contradictory) implications of nanomaterials/NPs in plants and other biota have also been reported (oxidative damage, interactions with various metabolic pathways by attacking membranes, lipids, DNA, and proteins, consequently reducing growth performance) [3,101,111]; thus, further elucidations of plant–nano-based–material interactions are needed due to the possible longer persistence and substantially lower detection of NPs in the environment. 

## 7. Environmental Interaction(s) and Additive Effects of Salinity-Exposed Plants

Salt-induced disorders in natural (agro)ecosystems are often accompanied by different additive environmental constraints, such as metal(s) contamination, water disbalance, Na-alkalinity, soil organic matter depletion, texture-light, and disaggregated pedosphere [8]. Multi-interactions among such numerous environmental variables result in multi-collinearity to crop responses, making it very complicated to study and elucidate their individual/combined impacts, making it almost impossible to detect which constraint (i.e., stress/factor) is the major limitation, e.g., [112]. Consequently, the management strategy (Figure 2) in saline areas can be extremely complicated, with a usually low cost-to-benefit ratio for crop food production. For instance, the authors of [69] recently studied controlled tripartite interactions (arbuscular mycorrhizal fungi (AMF), associated bacteria (AB), and salt-sensitive test crop maize) and observed that AMF treatment and AMF–AB interactions (vs. untreated control) alleviated the salt-induced reduction of test plant growth, root colonization, and nutrient accumulation, lowered leaf proline concentration, and finally improved the salinity resistance of maize through the dually induced effect exerted by AMF and AB. The same authors concluded that AB acted as an AMF helper and enhanced maize growth; however, AMF was likely the dominant player in the AMF–AB relationship under exposed salinity stress. In general, AMF (in addition to the abovementioned ameliorative impacts on salt-stressed plants) underpins the AB population in the rhizosphere [113], while vice versa, AB can stimulate the growth of AMF, which additionally underpins plant–microbe symbioses as a third factor, e.g., [114]. 

It was also documented that the interaction of salinity (NaCl) and metal (Cd) stress can enhance Cd phytoextraction as well, likely by exacerbating secondary oxidative disorders, i.e., increased plasma membrane permeability to nonessential Cd [115]. Furthermore, the authors of [10] recently confirmed several significant two-way interactions among soil NaCl salinity, Cd contamination, and/or humic acid (HA) addition on the chemical performance of rhizosphere solution (e.g., pH, metal speciation) and Cd soil–plant transfer in tissues of two radish cultivars. For example, in the rhizosphere solution, they noticed that: (i) the NaClxCd interaction (*p* < 0.008 for Sparkler cv., and *p* < 0.0001 for Cherry Belle cv.) was brought about by an increase in total Cd concentration with increasing NaCl salinity, while (ii) the CdxHA interaction (*p* < 0.01 for Sparkler cv., and *p* < 0.0009 for Cherry Belle cv.) resulted in a decrease in Cd concentration with increasing HA levels. With respect to Cd transfer to radish tissues, the same authors noticed that: (i) the NaClxCd interaction (*p* = 0.013 for Sparkler cv.) increased the Cd level in radish fruit (hypocotyl) with increasing NaCl addition, while (ii) the CdxHA interaction (*p* < 0.0001 for Cherry Belle cv.) reduced Cd levels in hypocotyl, notably at the highest Cd rate, with increasing HA. As previously elaborated with the speciation modeling approach (Section 2), metal-(in)organic complexation (e.g., among chlorides, complex organics such as humates, fulvates) is still not fully explained, albeit it is suggested that the biogeochemistry of deprotonated soil organics and dissolved chlorides are essential in driving metal bioavailability and soil–plant transfer. 

## 8. Exploration of Salinization Processes by Artificial Intelligence and Machine Learning Approaches

Artificial intelligence (AI) is a specific niche of computer science that uses algorithms and techniques trying to mimic human intelligent behavior. AI algorithms have the ability of self-learning without assuming parental distribution. They are extremely flexible and are used for different datasets in different application domains. The aim of AI is to create a model that is predictive and attempts to find the hidden patterns of a complex issue [116]. In parallel, machine learning (ML), as a type of AI, provides computers with the ability to learn without being explicitly programmed, using statistical methods to enable machines to improve with experience [117]. As a result, AI and ML effectively automate the process of analytical model-building and allow machines to generate new scenarios independently. For instance, some such widely applied models and algorithms are: (i) Artificial Neural Network (ANN, applied in almost all AI-based applications in real-life data) [116,118,119], (ii) Support Vector Machine (SVM, originally developed for linear classification problems) [120,121], (iii) Nonlinear Support Vector Regression (NLSVR), (iv) Random forest (RF) [122], and many others (Table 2). However, in some cases, ML (optimization) algorithms are employed to correct the assumptions of classical statistical models to obtain the robust predictions [123,124]. Additionally, for specific situations, neither linear nor nonlinear models provide better fitting; therefore, a combination of two or more algorithms/statistical models is needed, resulting in so-called hybrid or two-stage modeling approaches [116,125,126,127,128,129]. 

The AI and ML algorithms have been successfully applied in a wide range of agro-environmental areas, including: plant-based [124,129,130,131,132,133], pedological [134,135,136,137], and salinity-based [116,120,138] studies (Table 2).

However, as soil salinization is commonly a highly complex and nonlinear variable [12], the data processed by AI and ML techniques could result in better outcomes vs. classical statistical methods in soil salinity classification and prediction. Since there is no model that can be applied universally for every type of dataset, there will be as many situations where one ML technique will overcome another, and vice versa, e.g., [147], not permitting the assumption that any of the discussed models will perform better in all scenarios.

Only a few studies have been performed to address the challenges in measuring and modeling soil salinity, employing diverse statistical and AI approaches. For instance, the authors of [148] estimated EC by inputting several pedovariables (particle size distribution, CEC, drainage performances, organic matter, salinity) to develop a multilayer perceptron optimized with the Firefly algorithm (MLP-FFA) and evaluated its performance with the stand-alone MLP and ordinary kriging approaches. Interestingly, in the same study, MLP-FFA achieved higher accuracy vs. stand-alone MLP and ordinary kriging methods. The authors of [149] implemented SVM classifier models to categorize salt-affected soils with multispectral and texture properties as input variables. As a result, SVM was effective at extracting salinization and soil-thematic information from the inputs, leading to a valid classification status, while modeling soil salinity. Thus, in forecasting modeling, a substantial challenge confronted by AI and ML methods will need to find optimal weights in a neuronal layer to facilitate the extraction of appropriate input data as requirements for creating an optimal predictive model, e.g., [116].

## 9. Conclusions

Recent projections warn that salinization of soils and water resources is likely to increase as global climate change continues. Both of these natural resources are critical to crop food production and environmental and human health [2], and thus further pressures and negative implications (e.g., spreading of salt stress on croplands) under salinization of agroecosystems are among the very realistic scenarios for the near future. Regardless of salt-induced disorders (e.g., sodium/chloride-induced salt stress), each type of salinity (neutral/alkaline) should often be managed with integrative approach(es), which is one of the most difficult tasks for all stakeholders involved in agri-food production (farmers, scientists, industry, legislative representatives). Various preventive and proactive solutions have been applied over time in salt-stressed agroecosystems to improve plant nutrition, with clearly defined side effects. Some relatively new and modern approaches (e.g., the use of microbe–plant associations, application of nanotechnologies) have been shown to be very effective in alleviating various agricultural constraints, including salt stress, although in some cases the full results are still unknown or contradictory (e.g., negative effects on some biota). Multidisciplinary approaches and solutions driven not only by plant and agri-environmental scientists, but also those from other areas (remote sensing, artificial intelligence, machine learning, big data analyses), can ensure very useful tools for detecting, protecting, and controlling salinization, and thus minimizing the damage caused by salt stress.

## Figures and Tables

**Figure 1 plants-11-00717-f001:**
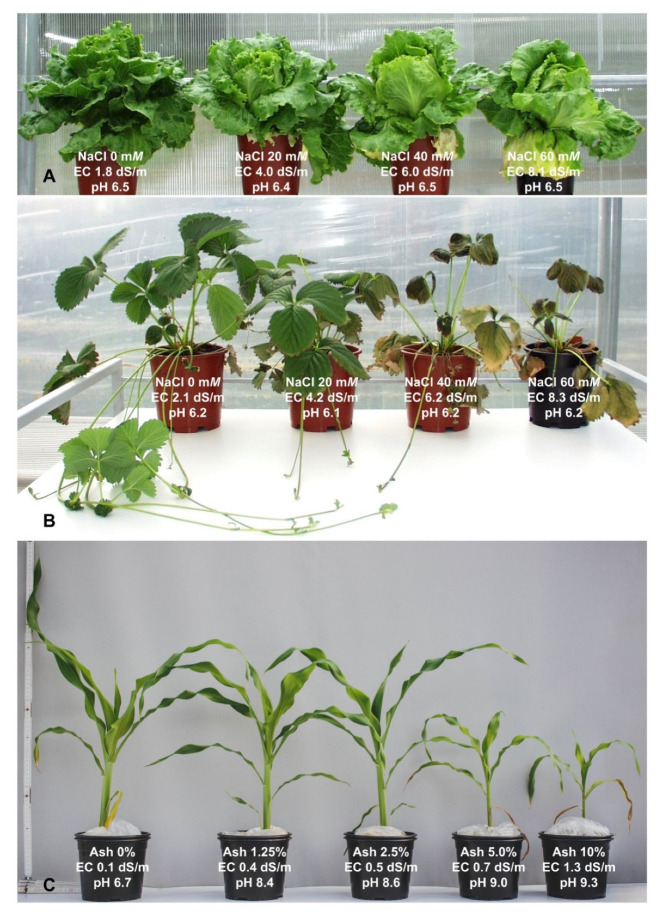
Vegetative and growth performances of lettuce (**A**) and strawberry (**B**) exposed to neutral salt stress (0–60 mM NaCl), and corn (**C**) exposed alkaline salt stress (0–10% wood-derived ash) with induced soil electrical conductivity (EC) and pH changes, after [12,26,27].

**Figure 2 plants-11-00717-f002:**
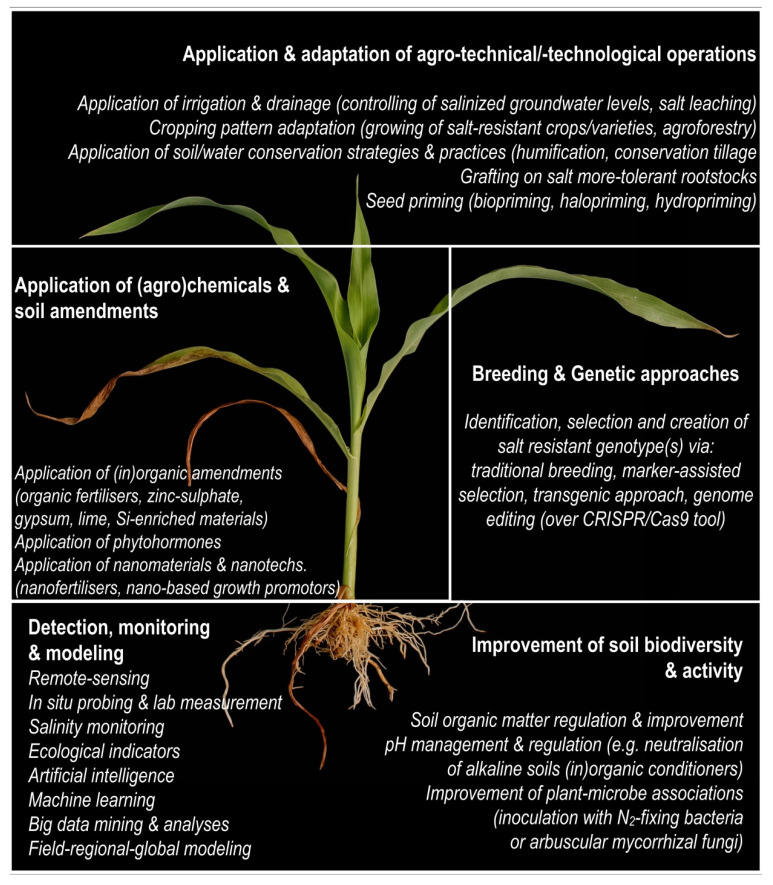
Solutions for salinization management and mitigating salt stress in plants.

**Table 1 plants-11-00717-t001:** Chemical speciation reactions for widely studied neutral and alkaline salt types dissolved in rhizosphere solution (after [12]).

Neutral Salt Type	pH	Prevalent Ions (%)	Precipitated Forms
Sodium chloride NaCl	7.94	Na^+^ 98; Cl^−^ 98	Magnesite Dolomite Hydroxyapatite Calcite Huntite Vaterite Artinite
Potassium chloride KCl	7.94	K^+^ 98; Cl^−^ 98	
Magnesium chloride Mg Cl_2_	7.93	Mg^2+^ 77; Cl^−^ 98; Mg-OC 10; MgSO_4_ 4; MgHCO_3_^+^ 4; MgCl^+^ 1	
Calcium chloride CaCl_2_	7.93	Ca^2+^ 78; Cl^−^ 98; Ca-organo-complexed forms 6; CaSO_4_ 6; CaHCO_3_^+^ 5; CaCl^+^ 1	
Sodium sulphate Na_2_SO_4_	7.94	Na^+^ 98; SO_4_^2−^ 72; CaSO_4_^−^ 16; MgSO_4_^−^ 10	
**Alkaline Salt Type**	**pH**	**Prevalent Ions (%)**	
Sodium hydrogencarbonate NaHCO_3_	8.01	Na^+^ 98; HCO_3_^−^ 92; CaHCO_3_^+^ 2; CaCO_3_ 1	
Sodium carbonate Na_2_CO_3_	8.08	Na^+^ 98.2; HCO_3_^−^ 92; CaHCO_3_^+^ 2.2; CaCO_3_ 1.3	
Potassium carbonate K_2_CO_3_	8.03	K^+^ 98; HCO_3_^−^ 92; CaHCO_3_^+^ 2; CaCO_3_ 1	
Magnesium carbonate MgCO_3_	8.07	Mg^2+^ 75; Mg-organo-complexed forms 10; MgHCO_3_^+^ 5; MgSO_4_ 4; HCO_3_^−^ 91; MgHCO_3_^+^ 2	
Calcium carbonate CaCO_3_	8.07	Ca^2+^ 76; Ca-organo-complexed forms 6; CaHCO_3_^+^ 6; CaSO_4_ 6; HCO_3_^−^ 90; CO_3_^2−^ 1	

**Table 2 plants-11-00717-t002:** Performance of AI and/or ML models in selected studies.

Area of Application	AI/ML Tools Applied	Best Performing Model	Reference
Soil resistance to penetration prediction Soil hydrological classification Digital soil mapping	ANN, SVM	SVM	[139]
	Soil Survey Data, KNN, SVM, Decision Trees (DT) Classification Bagged Ensembles and Tree Bagger	SVM	[122]
	Multiple linear regression (MLR), RF, SVR, ANN and k-nearest neighbors (KNN)	RF	[140]
Disinfection protocol in seed germination Soil moisture prediction	Generalized regression neural network (GRNN)	GRNN	[132]
	Extreme learning machine (ELM), RF, Ensemble empirical mode decomposition (EEMD)-ELM, EEMD-RF Complete ensemble empirical mode decomposition with adaptive noise (CEEMDAN)-ELM, CEEMDAN-RF	EEMD-ELM	[141]
Soil electrical conductivity prediction Soil salinity mapping Prediction of secondary compression index Soil nutrients prediction	Multilayer Perceptron (MLP) Neural Network, Hybrid MLP -grey wolf optimizer (GWO) model	Hybrid (MLP-GWO) Model	[116]
	SVM, ANN, RF	SVM	[120]
	Multi-gene genetic programming (MGGP) Particle swarm optimization (PSO), ANN, ANN-PSO	MGGP	[142]
	RF, Naïve Bayes (NB), SVM, ANN, DT, and Least Square SVM (LS-SVM)	LS-SVM and ANN	[143]
Soil organic carbon prediction Salt content prediction	ANN, SV, RF, MLR	RF	[144]
	Chemical detection method, visible-near-infrared spectroscopy, and two-dimensional deep learning (2D-DL)	2D-DL	[145]
Soil salinity prediction	Auto Encoder (AE), ANN, SVM, KNN, DT	AE-SVM	[146]
Soil salinity prediction and mapping	MLR, RF Regression, SVR	RF Regression	[121]

## Data Availability

Not applicable.

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
