# Peer review of "Salt Stress in Plants and Mitigation Approaches"

_plants, 2022, doi:10.3390/plants11060717_

Round 1

Reviewer 1 Report

The review manuscript addresses the issue of salinity which is a significant challenge in agriculture. By reducing crop yields, salinity would contribute to food shortage and insecurity. The information provided will be useful for both students and established researchers as more effort of research and development should be put on the study and mitigation of salinity.

The review paper by Ondrasek et al is overall well written and there are some points that needs review or a clarification as indicated below:

Line 59,  “known as salt stress, after a mid-term period of exposure [1,2,3].”  in this sentence it is not clear what is the “mid-term” of what

Line  182   consociation of forage and cereal plants

Line 193  “ …sugarcane farming which increases the midterm return along with the total profit [36].” It is not clear what are the  “midterm return”

Lines 309 to 311 “Due to plants developing antioxidant defense systems for coping with sa-309 linity stress of which were brought on by oxidative damage, is the causation of the increase in antioxidant enzymes that are eminent because of the activation of plant resistance mechanisms, of which the application of biochar will regulate the synthesis [57].   This sentence is long and not clear, consider rewriting

Line 477 “However, some of their forms pose quite low use efficiency (20–50%; [89])” Review this sentence

Line 525 “ Consequently, some nano-encapsulated fertilizer forms have been confirmed with better survival in inoculated soils together with their controlled and prolonged release (effective even 60 days after application) into the soil e.g. [93,110]”. Review and clarify sentence, the survival of what?

Author Response

Please find enclosed our revised version R1 of the Manuscript No.: plants-1610429R1

Title: Salt stress in plants & mitigation approaches

The manuscript R1 version has been checked and improved according to your comments and suggestions with active track change option. Our responses (red text) to the comments (black text) are presented in the attached file.

Reviewer 2 Report

the manuscript is an overview of the effects of salt stress on plants and possible solutions in agronomic management and in the use of new technologies. I must admit that the objective of the work at the end of the introduction is not explained and the reader struggles to perceive what the author wanted to stress, also considering that the title does not announce that it is a kind of review. It would have been useful, being a form of review,  that was evident from the title itself. On the salinity issue, several reviews have been developed and this adds small contributions to those already present. In particular, section 5-6. Section 8 it is difficult to understand what the message was, if not a mere revision of some works.

 The work is well written. I believe that the additional contribution, to what has already been done by others on salinity, it is section 5 and 6, although caution must be taken because to date clear open filed application at farm level by AMF e PGPB is not yet evident. Many positive results on AMF and PGPB still refer to controlled environments and pot tests. The real world of the open field is a whole other story, and in the end the reality.

In a somewhat uncertain spirit, I accept the work in its present form.

Author Response

(The authors gave the same response as above.)

Reviewer 3 Report

Very good review on salinity mitigation with most recent data. This review could serve master and PhD student in their researsh. congradulations

Author Response

(The authors gave the same response as above.)
